



# First High-Resolution Surface Spectral Clear-Sky Ultraviolet Radiation Dataset across China (1981–2023): Development, Validation, and Variability

Qinghai Qi[+,1], Yuting Tan[+,1], Christian A. Gueymard[2], Martin Wild[3], Bo Hu[4], Wenmin Qin[*,1], Taowen Sun[1], Ming Zhang[1], Lunche Wang[*,1]

[1]Hubei Key Laboratory of Regional Ecology and Environmental Change, China University of Geosciences, Wuhan, 430074, China

[2]Solar Consulting Services, Colebrook, NH, 03576, USA

[3]Institute for Atmospheric and Climate Science, ETH Zurich, Zurich, 8092, Switzerland

[4]State Key Laboratory of Atmospheric Boundary Layer Physics and Atmospheric Chemistry (LAPC), Institute of Atmospheric Physics, Chinese Academy of Sciences, Beijing, China

*Correspondence to*: Wenmin Qin (qinwenmin@cug.edu.cn), Lunche Wang (wang@cug.edu.cn)

**Abstract.** Solar ultraviolet radiation (UV) plays a fundamental role in the Earth's energy balance, influencing a wide range of processes, including material degradation, biophysical reactions, ecological dynamics, or public health. In this context, the first high-resolution (10×10 km) hourly dataset of surface solar UV under clear-sky conditions over mainland China from 1981 to 2023 is introduced, derived from ERA5 and MERRA2 reanalysis data and a reconstruction based on the SMARTS (Simple Model of the Atmospheric Radiative Transfer of Sunshine) spectral model. Leveraging the SMARTS model's accuracy and capabilities, this dataset provides UV data at 0.5 nm intervals between 280nm and 400nm, offering enhanced granularity for wavelength-specific analysis, thus filling a key gap in high-resolution hourly UV data for China. Validation of the UV dataset against ground observations at 37 stations of the Chinese Ecosystem Research Network (CERN) demonstrates strong performance, with a correlation coefficient (R), root mean square error (RMSE), and mean bias error (MBE) of 0.919, 5.07 W m$^{-2}$ and −0.07 W m$^{-2}$, respectively. Compared with the Earth's Radiant Energy System (CERES) UV product, this dataset offers higher spatial and temporal resolution as well as higher accuracy in comparison with observations, thus enhancing data quality for a wide range of applications. The spatial and temporal distribution of clear-sky UV radiation exhibits distinct regional and seasonal variations, with higher values in the west and south, and lower values in the east and north. Over the past 43 years, the annual mean clear-sky broadband UV radiation averaged over China was 20.05 W m$^{-2}$, showing a slightly increasing trend (+0.0237 W m$^{-2}$yr$^{-1}$). This dataset is now available at

https://cjgeodata.cug.edu.cn/#/pageDetail?id=110 or https://doi.org/10.6084/m9.figshare.28234298 , offering a valuable resource for addressing regional challenges related to UV radiation.

## 1 Introduction

Although ultraviolet (UV) radiation accounts for a small fraction (≈8%) of the total solar radiation (Gueymard, 2004), it significantly impacts human health, ecosystems, and various atmospheric photochemical reactions (Neale et al., 2023; Thomas et al., 2012), in particular. A moderate level of UV radiation facilitates the synthesis of vitamin D3 in the human body, essential for calcium absorption and bones health (Novotná et al., 2024; Wu et al., 2025). However, prolonged exposure to UV radiation can be harmful to the skin and eyes (Ma et al., 2023; Narayanan et al., 2010). UV radiation not only directly damages surface cells of plant leaves but also penetrates aquatic environment, affecting phytoplankton and weakening photosynthesis, which impacts plant growth (Williamson et al., 2014). UV radiation in the troposphere can accelerate photochemical reactions in the near-surface layer, contributing to the formation of secondary pollutants (Goti et al., 2024). Consequently, acquiring accurate UV data is essential in various fields, including public health, climate modeling, photobiology, ecosystem monitoring, and environmental management.

In the 1980s, the American Antarctic Program and the National Science Foundation established a high-latitude UV monitoring and observation network (Booth et al., 1994). In parallel, the Australian Radiation Laboratory (ARL) monitored UV radiation with spectroradiometers (SRM) and broadband detectors at 18 sites in Australia and Antarctica (Roy et al., 1998). In the 1990s, China set up its UV observation network with the establishment of a Brewer UV SRM at both the Zhongshan Research Station in Antarctica and the Waliguan Global Atmosphere Station on the Qinghai-Tibet Plateau, which marked the beginning of the UV observation network in China (Bo et al., 2009). The Chinese Ecosystem Research Network (CERN) established in 2004, continues to provide high-quality long-term UV data (Tong et al., 2023). Despite these efforts, the number of UV monitoring stations remains sparse compared to conventional meteorological or radiometric stations, leaving many regions of China uncovered (Lu et al., 2023). To address this gap, various methods for UV estimation have been developed, primarily classified into station-based and satellite-based approaches (Tang et al., 2022).

Station-based methods estimate the UV irradiance through an empirical model that ingests local



meteorological and/or radiometric data (Laiwarin et al., 2023; Qin et al., 2020b). For instance, the
normalized global surface irradiance (usually referred to as "clearness index"), solar zenith angle, and
total ozone column data can be used to estimate UV radiation (Antón et al., 2011; Wang et al., 2013).
These methods have been applied successfully in regions such as the Tibetan Plateau, the Pearl River
Delta and the North China Plain in China (Xia et al., 2008; Gong et al., 2014; Peng et al., 2015). Another
alternative method is to utilize the locally measured global solar irradiance and other meteorological
variables as inputs to empirical models (Barbero et al., 2006; Habte et al., 2019). Whereas station-based
methods are expected to provide accurate UV estimates near the station under scrutiny, their applicability
is limited when attempting to generalize results to other regions. Furthermore, these methods are sensitive
to local weather conditions and geographic variations, making them unfit to represent broader regional
patterns (Qi et al., 2024). One solution is to use gridded input data derived from reanalysis models or
from satellite observations. In particular, satellite-based methods (Jesus et al., 2023) can map large-scale,
spatially continuous UV radiation based on observations from instruments like the Total Ozone Mapping
Spectrometer (TOMS) (Chubarova et al., 2020; Zerefos et al., 2023), the Global Ozone Monitoring
Experiment (GOME) (Kujanpää and Kalakoski, 2015; Parisi et al., 2021), the Ozone Monitoring
Instrument (OMI) (Valappil et al., 2024; Zhang et al., 2019), the TROPOspheric Monitoring Instrument
(TROPOMI) (Lakkala et al., 2020; Lamy et al., 2021), or the Fengyun-4 (FY-4) (Qin et al., 2023; Wang
et al., 2024). Several algorithms have been developed to estimate UV from satellite data, including
statistical methods (Katsambas et al., 1997; Laguarda and Abal, 2019; Pei and He, 2019), look-up table
(LUT) methods (Leng et al., 2023; Su et al., 2005; Verdebout, 2000), and radiative transfer models (Janjai
et al., 2010).

Radiative transfer models explicitly consider various atmospheric processes such as scattering and
absorption from ozone, air molecules, clouds, or aerosols (Huang et al., 2019). For instance, Meerkoetter
et al. (1997) and Li et al. (2000) calculate UV using models based on the matrix-operator theory and the
discrete ordinate method (DISORT), respectively. Both methods account for multiple scattering and
absorption processes. Such models have been proven effective for UV radiation estimation, but are
computationally intensive and sensitive to various uncertainties, such as cloud fractional cover or adverse
weather conditions, making them less suitable for large-scale UV estimation over extended spatial and
temporal scales (Wu et al., 2022a, 2024).

In the past, many UV datasets containing long-term time series have been reconstructed based on satellite data (Ciren and Li, 2003; Čížková et al., 2018; Fragkos et al., 2024; Pei and He, 2019). Over China, Liu et al. (2017) reconstructed daily UV data from 1961 to 2014 using an all-sky estimation model combined with a hybrid model partly based on satellite observations. However, the main model's inputs use station-based meteorological data, resulting in an incomplete spatial coverage and a low spatiotemporal resolution. An attractive alternative consists in using gridded reanalysis data because of their global coverage, consistency, and perfect spatiotemporal continuity (no data breaks, contrary to satellite-based data). Reanalyses of interest include ERA5 (Li et al., 2023; Xia et al., 2021), MERRA2 (Laguarda et al., 2024; Lipponen et al., 2020), JRA-55 (Japanese 55-year Reanalysis) (Krizan, 2024; Wang et al., 2019) and CFSR (Climate Forecast System Reanalysis) (Li et al., 2024; Wang et al., 2011). Qin et al. (2020a) utilized MERRA2 to establish a novel physical broadband parameterization (FASTUV) for estimating surface solar UV radiation under all-sky conditions. Wu et al. (2022b) incorporated surface pressure and surface solar radiation from the ERA5 reanalysis data as input variables into their clear-sky UV radiation estimation model. Jiang et al. (2024) also used various ERA5 predictions, along with machine-learning methods, to derive the all-sky UV radiation over China on a daily basis.

Compared to the conventional methods reviewed above that only provide the *broadband* UV, the novelty of this study's goal is to obtain gridded estimates of the *spectral* UV irradiance under clear-sky conditions, and its spatiotemporal variations over mainland China. The remainder of this article is structured as follows. The methodology for estimating clear-sky UV radiation is presented in Section 2. Section 3 describes the input data used for that task, in addition to the observations and satellite data used for validation. Section 4 presents the validation results for the clear-sky UV radiation data, along with an analysis of the model's sensitivity and its spatial distribution over China. Section 5 discusses data availability, and finally Section 6 provides a summary and conclusions.

## 2 Methodologys

### 2.1 The SMARTS model

The "Simple Model for Atmospheric Transmission of Sunshine", or SMARTS for short, is a radiative transfer spectral model that has been under development since the early 1990s (Gueymard, 2001, 2005, 2019). SMARTS estimates the different clear-sky irradiance components (direct, diffuse, and

global) at the surface over the entire short-wave solar spectrum (280–4000 nm), and has become an essential tool in various disciplines, most importantly in various solar energy applications (Bicer et al., 2022; Mouhib et al., 2022; Pelland and Gueymard, 2022), the development of spectral irradiance standards (Habte et al., 2020; Xue and Igari, 2023), and UV research (Habte et al., 2019; N.Apell and McNeill, 2019). The UV band under scrutiny here extends from 280 to 400 nm at 0.5-nm resolution.

The SMARTS computational flow is shown in Figure 1. As visualized, single-image metadata is converted from planar to 3D at run time, thus significantly increasing the time cost of computation. To optimize the process, the model's Fortran code was reconstructed and converted to MATLAB®. In that optimized implementation, the model can perform matrix operations in an efficient way.

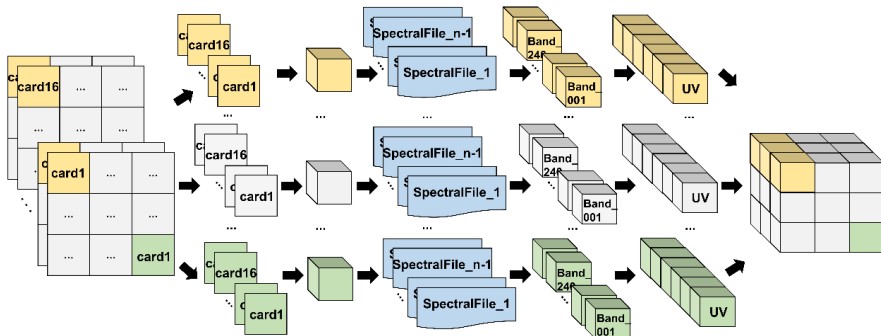

**Figure 1. Schematic diagram of the SMARTS workflow process for the large-scale application involved here.**

**2.2 Quality control of UV radiation observations**

The quality control of the measured UV data involves two parallel procedures: (i) quality control of the observations; and (ii) detection of the clear-sky periods in the observational time series (Figure 2). The quality control of observation data can be divided into two parts: detection and elimination of systematic errors, and quality assessment of the observation data. The systematic errors includes operator errors and sensor errors. Furthermore, many obvious errors and missing observations need to be eliminated. The quality assessment of observational data is conducted in successive steps. First, the observed surface broadband UV irradiance should be less than extraterrestrial UV radiation. Second, the ratio of UV radiation to total solar radiation should be in the range 0.02–0.08 (Liu et al., 2017).

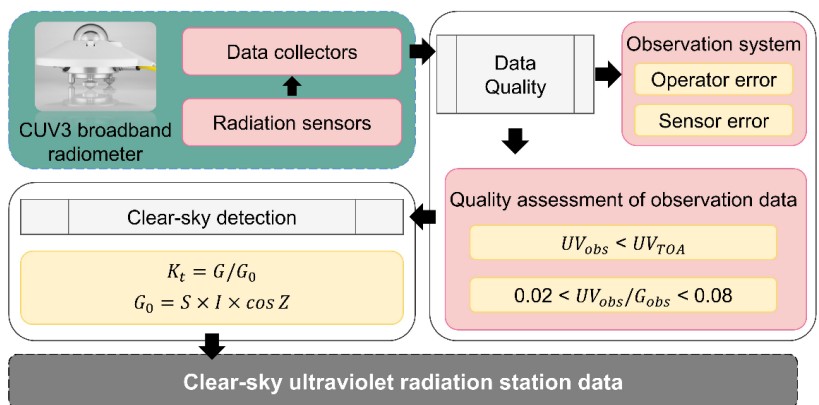

**Figure 2. Flowchart describing the quality control procedure of UV radiation observational data.**

It should be noted that, in the present investigation, the estimated UV radiation exclusively relates to clear-sky conditions. Therefore, it is necessary to conduct an efficient clear-sky screening of the observational time series, which needs to be done. The clearness index ($K_t$), which is the ratio between the surface global solar irradiance (G) measured at 1 hourly resolution and its extraterrestrial counterpart ($G_0$), is employed as the primary screening tool in this study. Clear-sky conditions are identified as those for which $K_t$ exceeds 0.7 (Qi et al., 2024). The global solar irradiance was obtained from the CERN site observations at 1 hourly temporal resolution and the extraterrestrial radiation was calculated using:

$$G_0 = S \times I \times \cos Z \tag{1}$$

where $S$ is the sun-earth distance correction factor, $I$ is the solar constant (1361.1 W m$^{-2}$) (Gueymard, 2018), and $Z$ is the solar zenith angle. At each instant, $Z$ and $s$ are provided by a precise sun position algorithm.

**2.3 Evaluation of the model**

In the absence of spectral measurements, which would necessitate costly SRMs, the assessment of the accuracy of the clear-sky UV modeled estimates is done relatively to broadband measurements, as described in Section 3.1. The assessment uses conventional evaluation metrics (Gueymard, 2014), including mean bias error (MBE), root mean square error (RMSE), and correlation coefficient (R). MBE and RMSE are expressed both in absolute terms (W m$^{-2}$) and relative percentage.

In addition, the probability density functions (PDFs) of the bias and the cumulative distribution functions (CDFs) of the absolute relative error (ARE) are used to compare the accuracy of the estimates



with respect to various satellite products (Jiang et al., 2020). The PDFs and CDFs provide complementary information about the concentration and distribution of the deviations between different UV radiation products and station observations. In particular, the position and shape of the PDF peaks can offer insights

into the temporal characteristics of the bias. The CDF curves of different UV radiation products also allow a visual comparison of their overall performance. The steeper the CDF curve, the more concentrated the error and the higher the accuracy of the product, whereas a flat curve indicates a dispersed error and lower precision.

**3 Data**

**3.1 Observational data from CERN**

The UV database collected from the China Ecosystem Research Network (CERN) observation sites is used to validate the accuracy of the UV radiation estimates obtained with the SMARTS model. CERN has established a network of ecosystem observatories throughout China to monitor the meteorological, environmental, and radiometric conditions over a widely diverse range of ecosystems, including nature

reserves, forest ecosystems, grassland ecosystems, wetland ecosystems, and urban ecosystems. The observational database selected for this study comprises hourly UV and total solar irradiance data from all the 37 stations in mainland China over the 2005-2013 period. The CERN stations use the CM11 global radiometer (Kipp & Zonen, The Netherlands) to measure global radiation with a spectral range of 285-2800 nm, with an accuracy of 5%. The CUV3 broadband radiometer (Kipp & Zonen, The Netherlands)

is used to measure UV radiation, with a spectral range of 280-400 nm and an accuracy of 5%, which meets the World Meteorological Organization's (WMO) measurement standards. As shown in Figure 3, the CERN sites are relatively well distributed in the east. Due to the influence of topography, however, the density of sites in the west is much lower than in the east, which results in an uneven spatial distribution overall.

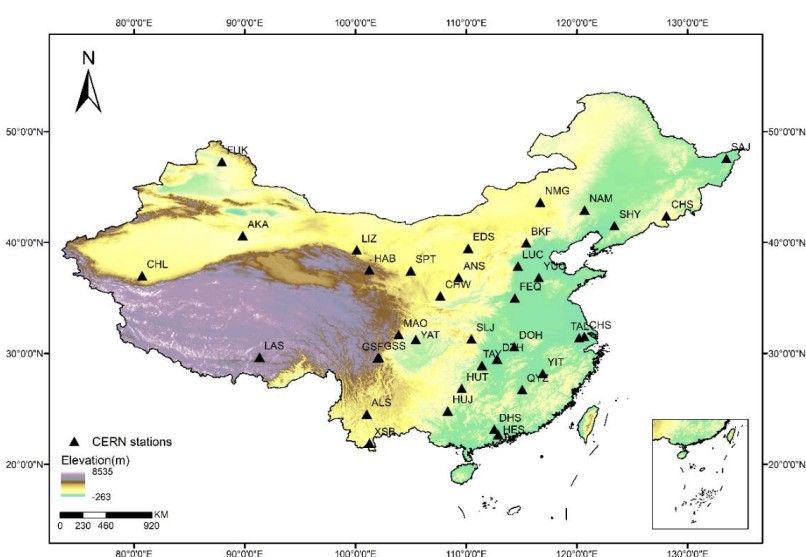


**Figure 3. Distribution of the 37 CERN sites in China that provide solar and UV irradiance observations.**

**3.2 Reanalysis-derived SMARTS model input variables**

The inputs to SMARTS are derived primarily from key products generated by the ERA5 and MERRA2 reanalysis over the period 1981–2023. ERA5 (ECMWF Reanalysis: Fifth generation) is an

atmospheric reanalysis dataset published by the European Centre for Medium-Range Weather Forecasts (ECMWF) to describe the state and variability of the Earth's atmospheric system(Muñoz-Sabater et al., 2021). The data employed in this study comprise ERA5_Land, ERA5_Pressure levels, and ERA5_Single levels. ERA5_Land has a spatial resolution of 0.1° × 0.1°, whereas ERA5_Pressure levels and ERA5_Single levels have a spatial resolution of 0.25° × 0.25°. Consequently, that coarser grid is

resampled to achieve a unified spatial resolution of 0.1° × 0.1° or ≈10 km.

MERRA-2 (Modern-Era Retrospective analysis for Research and Applications, Version 2) is an atmospheric reanalysis dataset released by NASA (Gelaro et al., 2017). MERRA-2 provides a long time series of AOD data (1980–present) with reasonable accuracy (Gueymard and Yang, 2020; Ou et al., 2022). The coarse resolution (0.5°×0.625°) of the MERRA2 AOD data requires bilinear resampling to 0.1° ×

0.1°, however. Table 1 shows the summarized information pertaining to all input variables. The position of each grid cell (latitude, longitude, and elevation) is also necessary to operate the radiation model.

**Table 1. Summarized information about the gridded variables used as inputs in this study.**



| Dataset name | Parameters name | Resolution | |
| --- | --- | --- | --- |
| | | Spatial | Temporal |
| ERA5-Pressure levels | Relative humidity (RH) | 0.25°×0.25° | Hourly |
| ERA5-Single levels | Total column ozone (TCO3), | 0.25°×0.25° | Hourly |
| | Total column water vapor (TCWV) | | |
| ERA5-Land | Surface pressure (SP), | 0.1°×0.1° | Hourly |
| | Forecast albedo (FAL), | | |
| | 2m temperature (T2M) | | |
| MERRA-2 | Aerosol optical depth (AOD) | 0.5°×0.625° | Hourly |

### 3.3 Satellite-based solar UV radiation product

CERES (Clouds and the Earth's Radiant Energy System) is a project initiated by NASA with the objective of conducting long-term monitoring and research on the Earth's radiant energy balance. In particular, CERES SYN1deg is a synthetic data base that provides various products related to the surface radiant field including radiative fluxes, cloudiness, and temperature. The SYN1deg database provides global coverage at a horizontal resolution of $1° × 1°$, with hourly temporal resolution (Wielicki et al., 1998). Here, the hourly UVA and UVB estimates from SYN1deg are used as a benchmark against which the present model's estimates can be assessed.

### 4 Results and discussion

### 4.1 SMARTS-derived UV irradiance performance assessment

Figure 4 presents the validation results of the hourly clear-sky UV radiation model estimates, compared with 196,170 observed data points from 37 CERN stations during 2005–2013, after applying the clear-sky screening procedure described in Section 2.2. Overall, the estimated hourly clear-sky broadband UV irradiance exhibits satisfactory accuracy, as evidenced by Figure 4a, with an R of 0.919, an RMSE of 5.07 W m$^{-2}$, and an MBE of $-0.07$ W m$^{-2}$. The density scatter plot shows that the model performs well across most UV radiation levels, particularly in the range of 20–40 W m$^{-2}$, where the highest density of points is observed. However, at lower (<10 W m$^{-2}$) and higher (>50 W m$^{-2}$) UV radiation levels, the scatter points become more dispersed, indicating increased model uncertainty under these extreme conditions. Notably, the regression slope of 1.009 and a negative intercept ($-0.319$)



suggest a slight systematic underestimation under low UV radiation conditions, possibly due to factors like local atmospheric characteristics not fully captured by the model. In addition, the upper and lower boundaries of the scatter points reveal distinct patterns: overestimation occurs predominantly under high

UV conditions, whereas underestimation is more frequent at low to moderate radiation levels. The broader scatter at extreme values also suggests that the model's performance might be sensitive to solar zenith angles, particularly during sunrise and sunset.

In addition, boxplots of R, MBE and RMSE for all stations combines are shown in Figure 4b. In particular, the RMSE for the majority of stations ranges between 3.9 W m$^{-2}$ and 5.6 W m$^{-2}$, with a mean

value of 4.98 W m$^{-2}$ and a maximum value not exceeding 9 W m$^{-2}$. The mean and median MBE values are −0.13 W m$^{-2}$ and 0.27 W m$^{-2}$, respectively, indicating a small overall error. The R-value at most sites is greater than 0.9. The median value is notably elevated, approaching the upper quartile, which collectively suggests that the overall R-value remains satisfactory and that the model's performance has strong spatial consistency and robustness, even in regions with varying climatic conditions. Figures 4c

and 4d display the RMSE and MBE results, respectively, for each station. About half of the stations are affected by only low RMSEs below 5 W m$^{-2}$. The two most challenging stations are Taihu (TAL) and Aksu (AKA). The Taihu (TAL) station, located in the humid, cloud-prone region of eastern China, exhibits significant errors, with RMSE and MBE reach 8.82 W m$^{-2}$ and −7.59 W m$^{-2}$, respectively. These errors are likely influenced by high humidity and the reflective surface of the Taihu Lake, which

complicates the estimation of clear-sky UV radiation. The Aksu (AKA) station, situated in the arid desert region of northwestern China, also stands out with RMSE = 6.40 W m$^{-2}$, MBE = −3.09 W m$^{-2}$, and R = 0.877. The larger errors at AKA may be attributed to several factors, including the high aerosol loading in the region, where mineral dust particles from the surrounding deserts can significantly scatter and absorb UV radiation. Additionally, the reflective properties of the desert surface, extreme climatic

conditions, and local meteorological variations could further complicate the accurate estimation of clear-sky UV radiation, leading to the observed discrepancies. These unique atmospheric conditions, combined with the station's extreme climatic environment, likely reduce the model's estimation accuracy. (Wu et al., 2022b).



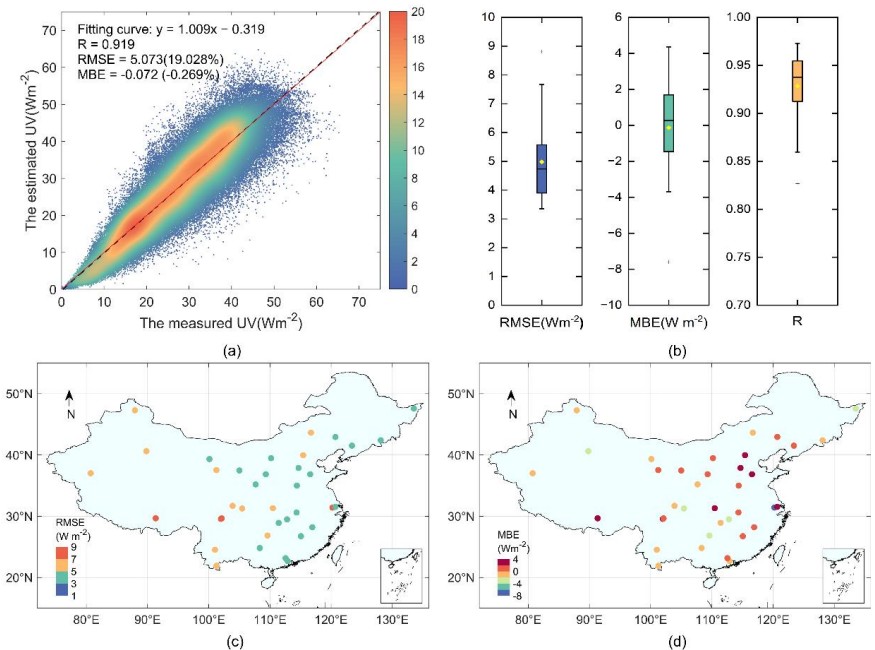

**Figure 4. Validation results of the estimated hourly clear-sky UV dataset against observations at 37 CERN stations from 2005-2013. (a) Density scatter plot between the estimated dataset and observations. (b) Box plots of the three statistical error metrics (RMSE, MBE, and R). (c) Spatial distribution of RMSE at individual stations. (d) Spatial distribution of MBE at individual stations.**

Figure 5 presents similar results as in Figure 4, but on a daily mean basis. From Figure 5a, the overall validation results are satisfactory, with an R of 0.907, an RMSE of 3.37 W m$^{-2}$, and an MBE of 0.14 W m$^{-2}$. The box plots in Figure 5b show that most stations have RMSE values below 4 W m$^{-2}$, reflecting consistent performance across regions. The MBE values are tightly clustered around zero, indicating minimal biases overall. Additionally, R-values for most stations range between 0.88 and 0.95, demonstrating the model's strong spatial consistency and reliable performance. The spatial distributions of RMSE and MBE in Figure 5c and 5d reveal patterns similar to those observed with the hourly estimates. Whereas most stations maintain low RMSE and near-zero MBE, the same two stations as before, TAL and AKA exhibit notable discrepancies again. For instance, the daily-mean RMSE, MBE, and R at AKA are 7.07 W m$^{-2}$, −6.16 W m$^{-2}$, and 0.783, respectively.

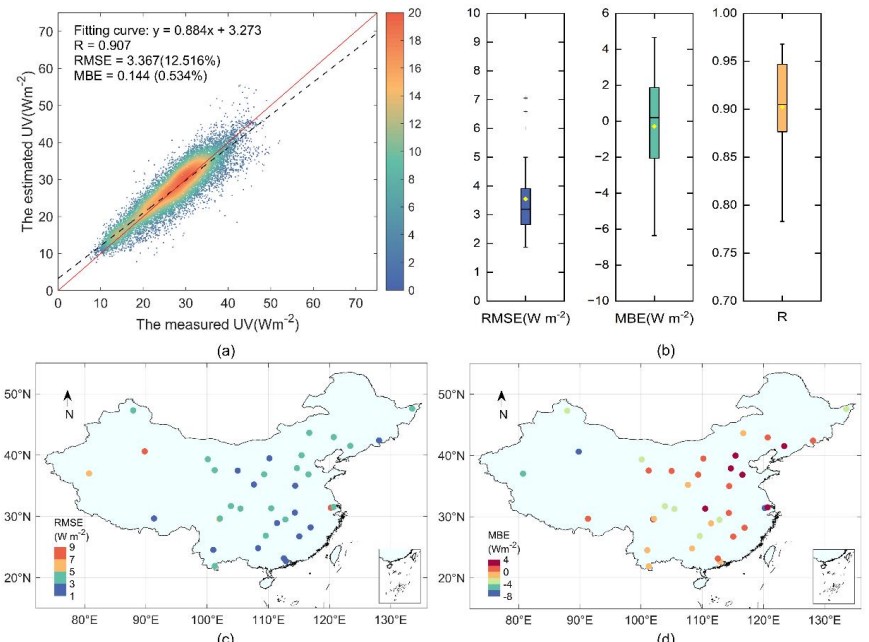

Figure 5. Same as **Figure 4** but for daily mean clear-sky UV radiation.

Overall, the validation results demonstrate that the model reliably captures hourly and daily mean clear-sky UV radiation across diverse environments, maintaining high accuracy and minimal bias at all 37 sites, with the possible exception of two challenging stations. The normally strong agreement between the model estimates and observations underscores its robustness. These findings support the reliability of the SMARTS model in estimating clear-sky UV radiation.

### 4.2 Benchmarking the estimated clear-sky UV radiation against the CERES product

A comparison between the present hourly UV estimates and those from CERES SYN1deg is desirable as a form of benchmarking, owing to the important status of the latter as a reference global database. Using CERN observations in 2013, the results in Figure 6 demonstrate that the present UV product significantly outperforms SYN1deg in terms of overall accuracy. As shown in Figures 6a and 6b, the estimated hourly clear-sky UV radiation achieves an R of 0.923, an RMSE of 4.92 W m$^{-2}$, and an MBE of 0.04 W m$^{-2}$, which are substantially better than the CERES product (R = 0.825, RMSE = 10.61 W m$^{-2}$, MBE = 4.68 W m$^{-2}$). Notably, SYN1deg exhibits systematic biases, with overestimation at high UV values and underestimation at low UV values, whereas the newly proposed product aligns more closely



with the observed values. The distribution of the PDF (Figure 6c) further highlights the differences

between the two products. The bias distribution of the newly proposed product closely resembles a

Gaussian curve, with a sharp peak near zero, indicating small and stable deviations across most data

points. In contrast, SYN1deg displays a broader and flatter PDF curve, suggesting larger and more widely

distributed errors, hence lower precision than the new SMARTS-derived product. Figure 6d reinforces

this conclusion: the CDF curve for ARE is steeper for the newly proposed product, reaching the 95%

threshold for a much lower mean error than SYN1deg. This confirms that a larger fraction of the

estimated product's deviations is confined to smaller bias ranges.

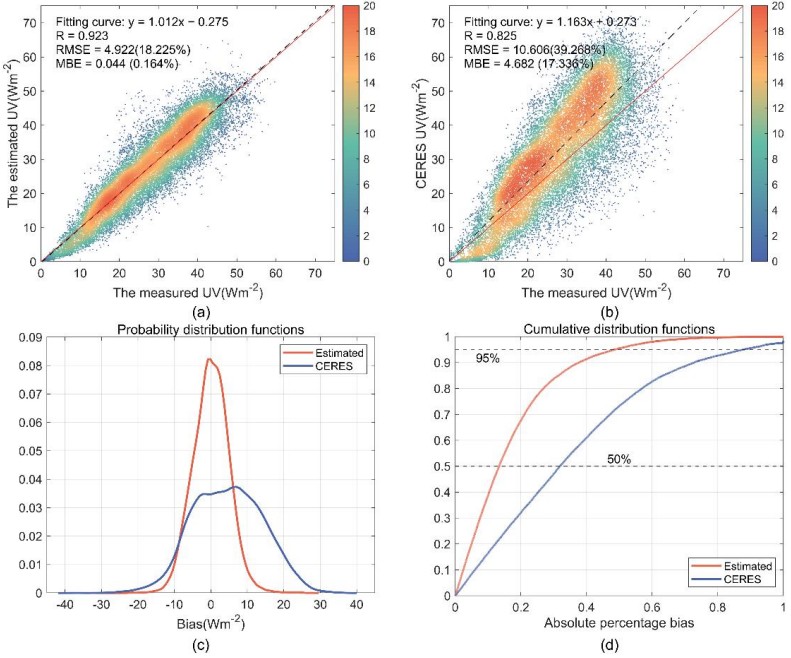

**Figure 6. Hourly assessment results for two clear-sky UV radiation products. (a) Density scatterplot between**

**the hourly clear-sky SMARTS-derived UV irradiance estimates and CERN observations. (b) Same as (a) but**

**for the CERES SYN1deg estimates. (c) Probability distribution function of the bias for the SMARTS-derived**

**product (orange line) and the CERES product (blue line); (d) Cumulative distribution function of the absolute**

**relative error for the SMARTS-derived product (orange line) and the CERES product (blue line).**

Figure 7 is similar to Figure 6, but on a coarser daily-mean resolution. The R value for the new UV

radiation product, 0.911, is markedly higher than the CERES product (0.763). In parallel, the RMSE and

MBE for the new product, 3.18 W m$^{-2}$ and 0.09 W m$^{-2}$, respectively, are much lower than their CERES

counterparts (8.84 W m$^{-2}$ and 6.39 W m$^{-2}$). From Figure 7b, it is evident that the CERES product

systematically overestimates UV radiation, as indicated by the positive deviation of its best-fit line from the 1:1 line in almost all cases. In contrast, the estimated product aligns much more closely with the observations. The PDF and CDF in Figures 7c and 7d further emphasize the differences between the two products, and confirm the results in Figures 6c and 6d.

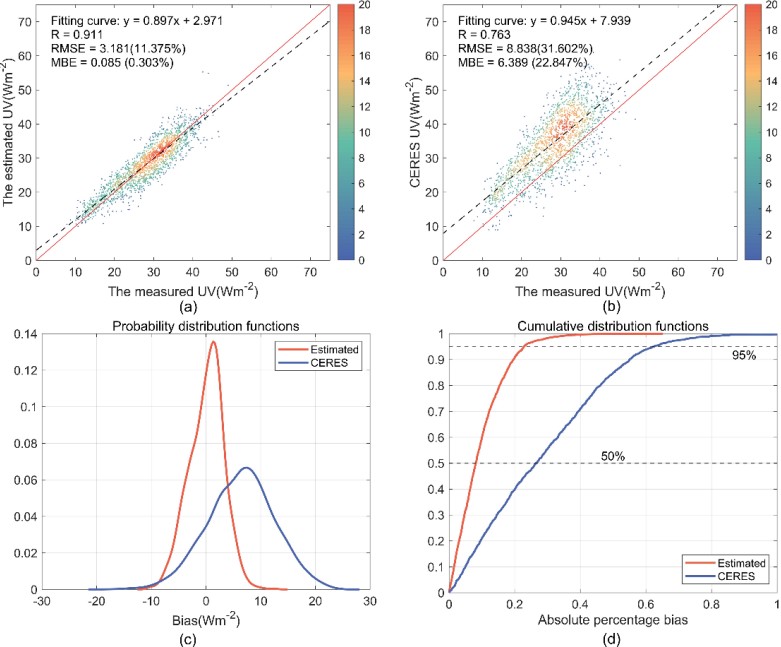

**Figure 7. Same as Figure 6 but for daily mean results.**

In summary, the hourly and daily mean clear-sky UV radiation product proposed here demonstrates significantly better performance than the CERES SYN1deg product in terms of accuracy, stability, and error distribution.

**4.3 Sensitivity of the SMARTS UV predictions to key input variables**

The results above indicate that the SMARTS-based UV estimates are satisfactory in general, but are nevertheless affected by reduced accuracy at a few sites where challenging situations are likely to impact the model's inputs. It is thus desirable to evaluate the sensitivity of the modeled UV estimates to discrepancies in the atmospheric inputs. To that effect, a series of scaling factors (0.5, 0.9, 1.1, and 1.5) are applied to three key input variables as way to analyze the intrinsic model's sensitivity to them. These inputs are the aerosol optical depth (AOD), the total column ozone (TCO3), and the forecast albedo

(FAL). In all cases, the clear-sky UV irradiances estimated using the original input data are used as the

baseline reference. As shown in Figure 8, the 37-station mean clear-sky UV irradiance thus obtained

exhibits a typical unimodal distribution, peaking in summer and reaching a minimum in winter, as could

be expected. This seasonal variation is closely tied to changes in solar zenith angle. Among the four input

parameters tested here, AOD induces the most significant impact on clear-sky UV radiation (Figure 8a).

As AOD increases, the UV irradiance decreases substantially, highlighting the strong attenuation effect

of aerosols in the UV (from -50.06% to -40.90% per AOD unit). This is consistent with the physical

properties of aerosols, which scatter and absorb radiation mostly in the UV spectrum. As TCO3 increases

incrementally relative to its original value, the clear-sky UV radiation decreases progressively (Figure

8b), which reflects the strong absorption of ozone in the UV spectrum, although this is confined to

wavelengths below ≈340 nm only. Surface albedo (FAL) is shown to enhance UV radiation when it

increases (Figure 8c). Higher albedo values, such as those from snow-covered surfaces, amplify the

reflection of UV radiation back into the atmosphere, which can subsequently increase the downward UV

irradiance through the atmospheric backscattering effect  (Shine et al., 2012).

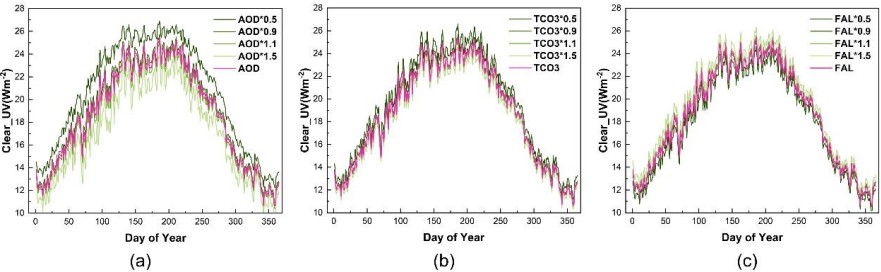

**Figure 8. Mean annual variation of estimated clear-sky UV irradiance at 37 CERN stations under different**
**conditions. The baseline (red curve) uses the original input data. The four green curves represent the estimates**
**obtained after scaling the indicated input.**

Figure 9 presents the validation results of the SMARTS model's clear-sky UV radiation estimates

against station observations under different scaling factors. For AOD, as the scaling factor increases from

0.5 to 1.5, the R value decreases from 0.943 to 0.918, indicating a slight decline in the model's correlation

with observations. The MBE shifts from 1.83 W m$^{-2}$ to −1.51 W m$^{-2}$, suggesting that the model

overestimates UV radiation at lower scaling factors and underestimates it at higher scaling factors. The

RMSE initially decreases from 5.08 W m$^{-2}$ to 4.87 W m$^{-2}$ (scaling factors 0.5–0.9), and subsequently

increases to 5.44 W m$^{-2}$ when the scaling factor exceeds 1, demonstrating a reduction in the model's

predictive capability with increasing AOD scaling factors. As the scaling factor for TCO3 increases, the

R value remains stable between 0.932 and 0.934, showing minimal variation. The MBE decreases from

1.19 W m$^{-2}$ to −0.74 W m$^{-2}$, indicating a transition from overestimation to underestimation. Meanwhile,

the RMSE decreases from 5.21 W m$^{-2}$ to 4.69 W m$^{-2}$, suggesting that the model's prediction error is

reduced at higher TCO3 scaling factors. When the FAL scaling factor reaches 1.5, the model exhibits the

lowest overall accuracy, with an R value of 0.929, MBE of 1.05 W m$^{-2}$, and RMSE of 5.29 W m$^{-2}$,

indicating that increasing FAL scaling factors significantly amplify the model's prediction error. The

MBE shifts from −0.98 W m$^{-2}$ to 1.05 W m$^{-2}$, demonstrating that the model underestimates at lower

scaling factors and overestimates at higher scaling factors. These results reveal the complex influence of

different input parameters on UV radiation estimation and provide critical insights for optimizing model

performance.

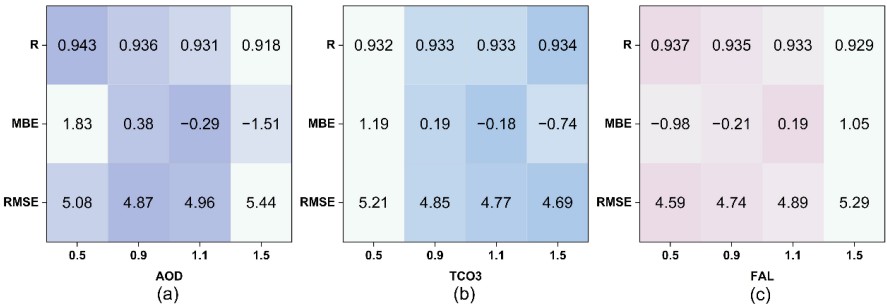


**Figure 9. Accuracy validation of clear-sky UV radiation estimates and station observations under different
conditions.**

### 4.4 Evaluating the spatial and temporal variability of clear-sky UV radiation

### 4.4.1 Evaluating the spatial variability of clear-sky UV radiation across wavelengths

In this study, we generated a stereogram of clear-sky UV radiation by wavelength using the

SMARTS model. The data range spans 280–400 nm with a step size of 0.5 nm, encompassing radiation

values across 241 wavelength bands (Figure 10a). The first layer of this figure illustrates the distribution

of total UV radiation under clear-sky conditions across China in 2023, while the subsequent 241 layers

represent the radiation value distributions for individual wavelength bands. Figure 10b shows an

unfolded representation of the mean clear-sky UV radiation values across China for each wavelength

bands from Figure 10a, showing a gradual increase in UV radiation intensity with increasing wavelength.

In the short wavelength range (280-310 nm), the radiation is relatively weak, likely due to strong

absorption by the ozone layer. In contrast, the long wavelength range (near 400 nm) exhibits stronger

radiation, indicating greater penetration capability. By analyzing the radiation intensity at different

wavelengths, the impact of UV radiation at various wavelengths on human health and the environment

can be studied.

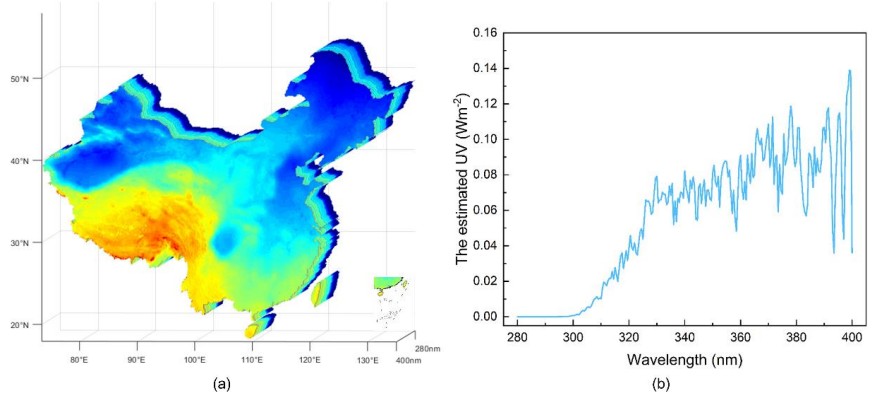

(a)    (b)

**Figure 10. (a) 3D hyperspectral distribution of clear-sky UV radiation in China (2023). (b) Spectral variation of clear-sky UV radiation (280–400 nm) across entire China.**

**4.4.2 Evaluating the spatial variability of clear-sky UV radiation**

The SMARTS model was employed to reconstruct a long-term dataset of clear-sky UV radiation

across China from 1981 to 2023. Figure 11 presents the annual average spatial distribution of clear-sky

UV radiation in China during 1981-2023.The clear-sky UV radiation values range from 14.26 W m$^{-2}$ to

31.25 W m$^{-2}$, with an overall annual average value of 20.05 W m$^{-2}$. Spatially, clear-sky UV radiation

exhibits a distinct west-to-east decreasing gradient, accompanied by lower values in northeastern China

compared to the southern regions. The Tibetan Plateau, characterized by its high altitude, exhibits the

highest clear-sky UV radiation levels in the country. The reduced atmospheric thickness over high-

altitude regions results in less attenuation of UV radiation, thereby enhancing surface UV levels. In

contrast, northeastern China experiences lower UV radiation due to its higher latitude, where lower solar

altitude angles lead to greater atmospheric absorption and scattering, significantly attenuating surface

UV radiation. Additionally, the Sichuan Basin is identified as a region with notably low clear-sky UV

radiation levels. This is attributed to its low elevation and surrounding mountains, which limits the

amount of solar radiation received, which enhances UV radiation absorption and scattering within the

atmosphere (Qin et al., 2020a).



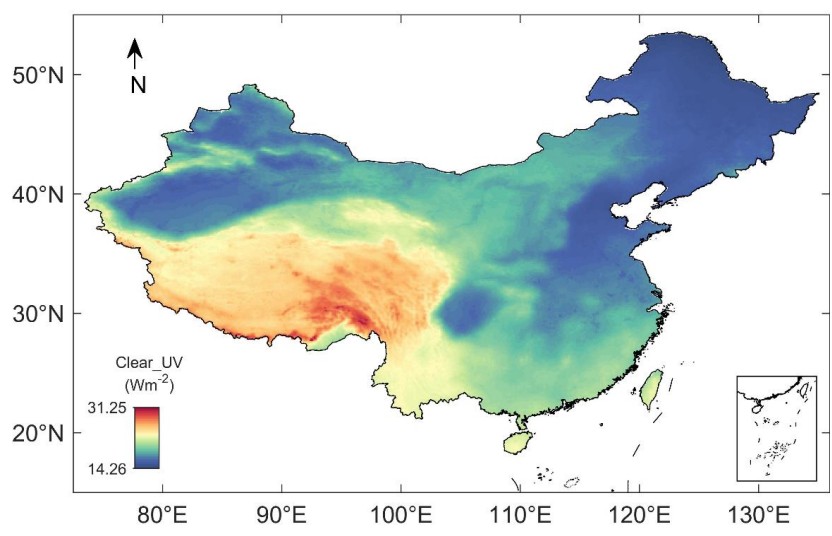

**Figure 11. Annual average spatial distribution of clear-sky UV radiation in China from 1981 to 2023.**

As illustrated in Figure 12, the clear-sky UV radiation in China exhibits pronounced seasonal variations, primarily driven by changes in the solar zenith angle and maximum sunshine duration throughout the year. The highest UV radiation levels are observed in summer (Figure 12b), particularly over the Tibetan Plateau, where values approach 35 W m$^{-2}$. Conversely, the winter (Figure 12d) demonstrate the least intensity of UV radiation, with values in northern regions reaching only 5-10 W m$^{-2}$. Spring (Figure 12a) and autumn (Figure 12c) exhibit intermediate radiation levels, reflecting the seasonal transition between the extremes of summer and winter. From a spatial perspective, the Tibetan Plateau consistently records the highest UV radiation levels across all seasons due to its considerable altitude and thinner atmosphere. In contrast, eastern regions of China, characterized by lower altitudes and denser atmospheric layers, display comparatively lower UV radiation levels. These spatial patterns underscore the influence of both altitude and latitude on the distribution of UV radiation across China.



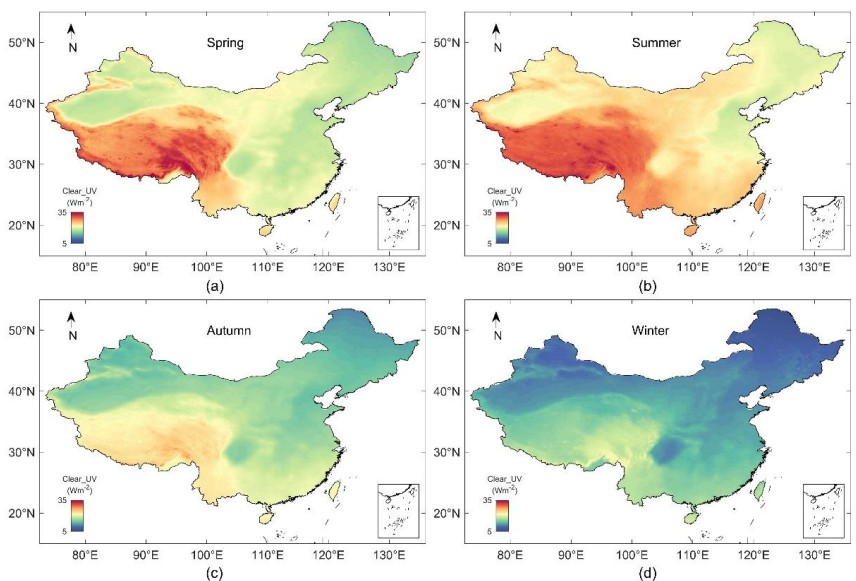

**Figure 12. Average seasonal spatial distribution of clear-sky UV radiation in China from 1981 to 2023.**

### 4.4.3 Evaluating the inter-annual variability and trends of clear-sky UV radiation

The annual trend of clear-sky UV radiation in the Chinese region from 1981 to 2023 is shown in Figure 13. Over the 43-year period, the annual average clear-sky UV radiation demonstrates a slight overall upward trend (+0.0237 W m$^{-2}$ yr$^{-1}$), with annual mean values ranging from 16.02 W m$^{-2}$ to 21.33 W m$^{-2}$. The inter-annual variability can be divided into three distinct phases. From 1981 to 1994, clear-sky UV radiation exhibited a decreasing trend (−0.0488 W m$^{-2}$ yr$^{-1}$). This reduction is primarily attributed to the eruption of the El Chichón volcano in Mexico in 1982 and the Mount Pinatubo eruption in the Philippines in 1991. A large amount of smoke and volcanic ash was released from the stratosphere into the troposphere, leading to a sharp increase in aerosol particle concentration in the atmosphere. High concentrations of aerosols significantly reduce the intensity of UV radiation reaching the Earth's surface through scattering and absorption processes. (Fang et al., 2021). In 1992, clear-sky UV radiation reached its lowest value in 43 years at 16.02 W m$^{-2}$. After 1992, the impact of volcanic eruptions on clear-sky UV radiation gradually diminished, with the overall trend levelling off by 1995. Following this period, from 1995 to 2009, with rapid economic development and the extensive burning of fossil fuels, air pollution became more severe, leading to an increase in anthropogenic aerosols. At the same time, the rise in ozone concentration over China enhanced the scattering and absorption processes in the





atmosphere (Verstraeten et al., 2015), thereby reducing the UV radiation reaching the Earth's surface ($-0.0329$ W m$^{-2}$ yr$^{-1}$). Since 2010, clear-sky UV radiation has risen sharply ($+0.0670$ W m$^{-2}$ yr$^{-1}$), coinciding with the implementation of stringent air pollution control policies in China, which effectively reducing the concentration of aerosols, ozone, and other pollutants in the atmosphere by curbing coal

consumption and industrial emissions (He et al., 2016). The continuous improvement in air quality has significantly enhanced the intensity of UV radiation under clear-sky conditions, showing a significant upward trend.

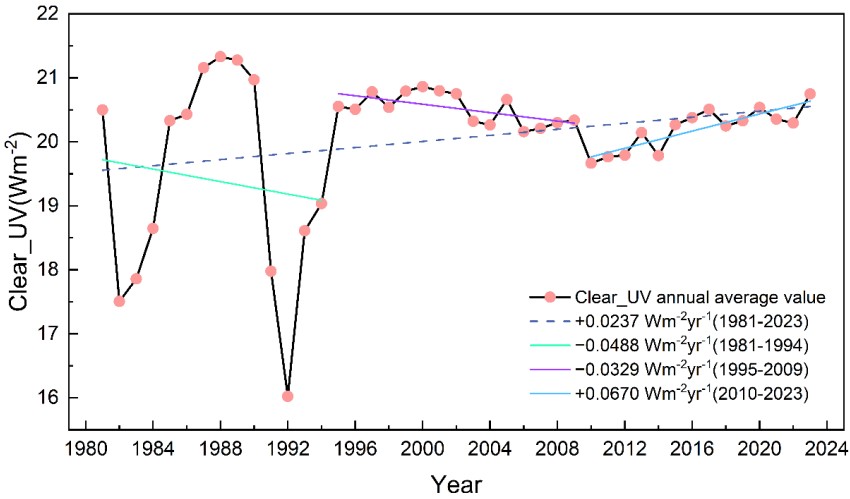

**Figure 13. Inter-annual variation trend of clear-sky UV radiation in China, 1981-2023.**

**5 Data availability**

The hourly clear-sky UV radiation dataset reconstructed in this study, covering 43 years (1981–2023), has been uploaded to the Middle Yangtze River Geoscience Data Center and figshare, and is stored in NetCDF format. Users can access the complete dataset via the links: https://cjgeodata.cug.edu.cn/#/pageDetail?id=110 or https://doi.org/10.6084/m9.figshare.28234298 (Qi

et al., 2025). The one-year datasets include 365 or 366 NetCDF files, with file names following the format "UV_yyyymmdd", where "yyyy" represents the year, "mm" represents the month, and "dd" stands for the day. The unit of the data is W m$^{-2}$. Each file contains three variables: solar UV radiation, longitude, and latitude, with dimensions of 616 × 356 × 24. The dataset covers the region from 73.5°E to 135°E and 18°N to 53.5°N, with a spatial resolution of 0.1° (approximately 10 km). The data is presented in local

standard time (LST), corresponding to UTC+8 for China.



In addition, leveraging the spectral computation capabilities of the SMARTS model, the dataset has been further refined to provide UV radiation data at 0.5 nm intervals within the wavelength range of 280–400 nm, covering a total of 241 bands. This refined dataset has also been uploaded to the Middle Yangtze River Geoscience Data Center (https://cjgeodata.cug.edu.cn/#/pageDetail?id=110) and figshare

(https://doi.org/10.6084/m9.figshare.28234298) (Qi et al., 2025), accessible through their respective links for download. Each file is named using the format "UV_05nm_yyyymmddhh," where "yyyy" denotes the year, "mm" represents the month, "dd" stands for the day, and "hh" denotes the hour, with units in W m⁻². Each file contains four variables (solar UV radiation, longitude, latitude, and wavelength), with dimensions of 616×356×241.

**6 Summary and conclusions**

We used the SMARTS model to reconstruct a long-term (1981-2023) hourly clear-sky UV radiation dataset (10 km×10 km) in China. The SMARTS model is capable of generating wavelength-specific clear-sky UV radiation with a spectral resolution of 0.5 nm, covering the 280-400 nm range, which enhances its ability to analyze spectral UV characteristics. Key inputs to the model include conventional

meteorological products from ERA5 (e.g., relative humidity, total column ozone, total column water vapor, surface pressure, 2m temperature and forecast albedo) and aerosol properties from MERRA2. The generated clear-sky UV radiation dataset was rigorously validated against CERN site observations and further compared with UV radiation estimates from CERES products. For the hourly clear-sky UV radiation, the overall R, RMSE and MBE were 0.919, 5.07 W m⁻² and −0.07 W m⁻², respectively. On the

daily scale, these values were 0.907, 3.37 W m⁻² and 0.14 W m⁻², respectively. The generated clear-sky UV radiation dataset significantly outperforms the CERES product on both temporal scales.

In addition, this study conducted a sensitivity analysis on the key input parameters (AOD, TCO3, and FAL) of the SMARTS model using scaling factors of 0.5, 0.9, 1.1, and 1.5. The results indicate that AOD and TCO3 are the primary attenuating factors for UV radiation, while FAL has an enhancing factor.

Specifically, as the AOD scaling factor increases, the model's predictive capability significantly declines. At higher TCO3 scaling factors, the model's prediction error decreases. Conversely, increasing the FAL scaling factor substantially amplifies the model's prediction error. The spatial distribution of clear-sky UV radiation in China generally exhibits a pattern of "higher in the west and lower in the east, higher in





the south and lower in the north". In addition, there is significant seasonal variation. The annual mean

values of clear-sky UV radiation range from 14.26 W m$^{-2}$ to 31.25 W m$^{-2}$, with the overall annual mean

reaching 20.05 W m$^{-2}$. By analyzing the spectral UV radiation distribution, it was observed that the clear-

sky UV radiation increases with wavelength across the 280-400 nm range. Throughout the study period,

clear-sky UV radiation exhibited a slight increasing trend (+0.0237 W m$^{-2}$ yr$^{-1}$).

This study reconstructed a long-term, high-temporal, and high-spatial resolution dataset of clear-

sky UV radiation and further refined it into a spectral UV radiation dataset with 0.5 nm intervals. We

anticipate that this dataset will provide valuable support for research in fields such as human health and

ecological environments. However, as the SMARTS model only simulates UV radiation under clear-sky

conditions, future research should integrate the effects of cloud cover to develop an all-sky UV radiation

dataset. Furthermore, the scope of the study could be expanded from the national to the global scale,

enabling a more comprehensive spatial and temporal analysis.

**Appendix A: Additional figures**

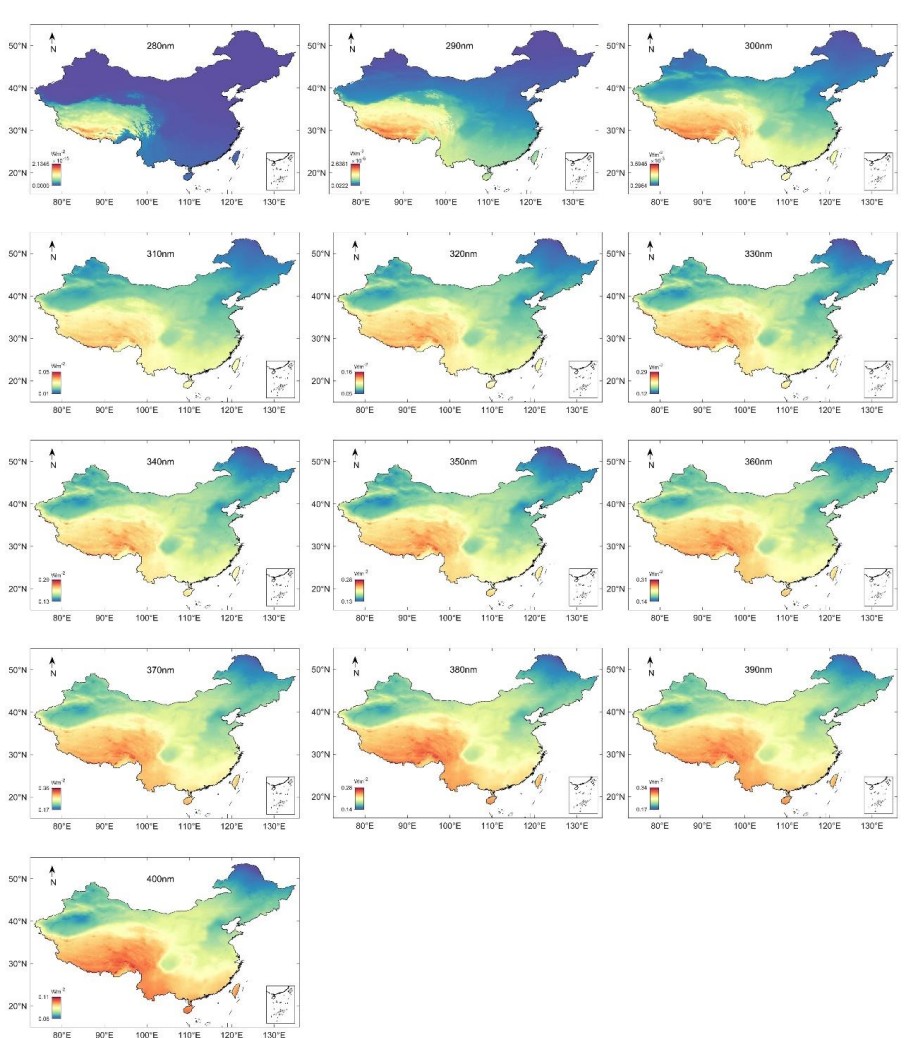

**Figure A1. Spatial distribution map of specific spectral bands in China (2023)**

**Author contributions**

QQ: designed the research, analyzed the results, wrote and revised the manuscript. YT: calculated the

dataset and validation. CAG: support the initial code of SMARTS model. WQ: designed the research,

revised the manuscript. HB: revised the manuscript and provide the CERN Data. TS: analyzed the data.

MZ: revised the manuscript. MW: revised the manuscript. LW: revised the manuscript.



**Competing interests**

The authors have the following competing interests: Martin Wild is a member of the editorial board of Earth System Science Data.

**Disclaimer**

Publisher's note: Copernicus Publications remains neutral with regard to jurisdictional claims made in the text, published maps, institutional affiliations, or any other geographical representation in this paper. 485 While Copernicus Publications makes every effort to include appropriate place names, the final responsibility lies with the authors.

**Acknowledgements**

The UV radiation station data were obtained from the China Ecosystem Research Network (CERN). The ERA5 conventional meteorological data and MERRA-2 aerosol data are available from their official 490 websites (https://cds.climate.copernicus.eu and https://disc.gsfc.nasa.gov). The authors would like to thank the staff of the data management and production organizations for their valuable work.

**Financial support**

This research was financially supported by the National Natural Science Foundation of China (No. 42371031).

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
