# Peer review of "First High-Resolution Surface Spectral Clear-Sky Ultraviolet Radiation Dataset across China (1981–2023): Development, Validation, and Variability"

_Earth System Science Data, 2025_

## Author Comment (AC1)

**Response to Referee #1**

First High-Resolution Surface Spectral Clear-Sky Ultraviolet Radiation Dataset across China (1981–2023): Development, Validation, and Variability

Qinghai Qi, Yuting Tan, Christian A. Gueymard, Martin Wild, Bo Hu, Wenmin Qin, Taowen Sun, Ming Zhang, Lunche Wang

29 November 2025

Dear editor and reviewers,

We would like to thank the editor for handling our manuscript and the reviewers for their careful evaluation of our work and the valuable comments, suggestions, and questions. Our point by point response to the comments made by Reviewers are given below, we have also marked changes in the manuscript. Please find a detailed point-by-point response to each comment.

Yours sincerely,

    Wenmin Qin, Qinghai Qi and co-authors

**General Comment:**

*The authors present a high-spectral-resolution clear-sky ultraviolet radiation dataset for China, which features unprecedented 0.5 nm spectral sampling across the 280-400 nm range. The integration of reanalysis data with physical radiative transfer model (SMARTS) represents a robust approach. Its credibility is further strengthened by comprehensive validation against ground-based CERN stations and CERES satellite products. The dataset appears to be a valuable resource, particularly given its spectral resolution and spatial coverage over China. The manuscript merits publication after addressing the following minor reversion.*

**Response:**

Thank you for your thorough and insightful review of our manuscript. We appreciate the detailed review and suggestion. We have carefully considered each of your comments.

**Specific Comments:**

*1) Abstract: Please correct "Earth's Radiant Energy System" to the full name "Clouds and the Earth's Radiant Energy System".*

**Response:**

Abstract has been changed as suggested.

*2) Section 2.2: The sun-earth distance correction factor have two different letters, "S" and "s"; standardise the format.*

**Response:**

The sun-earth distance correction factor has been unified as "S".

**3)** *Section 2.3: Phrase "concentration and distribution" is redundant, as the distribution characteristics of errors already encompass the concept of central tendency.*
**Response:**
We thank the reviewer for this insightful comment. As suggested, we have revised the manuscript by removing the redundant term "concentration and" and now simply refer to the "distribution" of errors throughout the text.

**4)** *Figure 3: Part of the site name is obscured in Figure 3.*
**Response:**
Thank you very much of your careful review of our research. We have replaced the Figure 3 with a new version.

**5)** *Table 1: How are the variables presented in Table 1 incorporated into the SMARTS model?*
**Response:**
We thank the reviewer for this insightful question regarding the role of the input parameters in the SMARTS model. In section 4.3, the discussion focused on three key input variables (aerosol , ozone and albedo) within the SMARTS model.

**6)** *Figure 4: The unites of solar radiation in Figure 4 are inconsistently represented, with both "Wm-2" and "W m-2" being used.*
**Response:**
As suggested, the unit of solar radiation is unified as "W m$^{-2}$". Meanwhile, the Figure 4 has been replaced with a new version.

**7)** *Figure 5: The same issue as in Figure 4.*
**Response:**
We have replaced the Figure 5 with a new version.

**8)** *Line 263: "The normally strong aggrement ..." is an inaccurate statement.*
**Response:**
The word "normally" has been removed to enhance the accuracy of the statement.

**9)** *Section 4.2: The conclusion could be enhanced by providing additional analytical insights regarding the distinctive features and relative performance of the two products, which would help users make informed selections.*
**Response:**
Thank you for pointing out the problem. We have added the statement in Section 4.2: "*The CERES SYN1deg product exhibits systematic biases in its representation of UV radiation, overestimating at high values and underestimating at low values. This results in a border and flatter probability density function compared to the new CHUV product, which indicates larger errors and lower overall precision. Consequently, the SYN1deg product also demonstrates a significantly weaker agreement with daily mean observations than the CHUV product.*"

**10)** *Line 315: "-50.06% to -40.90% per AOD unit" change in UV irradiance require clarification.*

**Response:**

We thank the reviewer for this astute observation and the opportunity to clarify this critical point. The original phrasing has been replaced with a more precise statement. "*The sensitivity analysis revealed a strong negative relationship between AOD and surface UV irradiance. Specifically, a one-unit increase in AOD was associated with a fractional reduction in UV irradiance ranging from -50.06% to -40.90%.*"

**11)** *Section 4.4.1: Replace the imprecise term "near 400 nm" with the scientifically accurate designation of either "UV-A range (315-400 nm)" or "long-wave interval (380-400 nm)".*

**Response:**

The suggested alterations have been implemented. It is evident that the statement "near 400 nm" has undergone a modification, with the numerical values now expressed as "380-400 nm".

**12)** *Section 4.4.2: The comparative analysis of the three regions would be strengthened by establishing a unified reference baseline, such as expressing their UV radiation levels as deviations from the national average value.*

**Response:**

We thank the reviewer for this suggestion. In our initial analysis, we utilized a national map of UV radiation to provide the spatial context, which indeed allows for a visual assessment of how the three regions compare against the national distribution.

**13)** *Figure 12: Please briefly quantify the radiation differences between transitional seasons (Spring/autumn).*

**Response:**

We thank the reviewer for this valuable suggestion. In the revised manuscript, we have added a quantitative analysis in Section 4.4.2: "*The average national UV radiation levels experienced a notable increase of 0.8962 W m$^{-2}$ in the spring months when compared to the annual mean. Conversely, during the autumn season, a decline of approximately 0.9277 W m$^{-2}$ was observed.*"

**14)** *Line 389: "considerable altitude" is not appropriate.*

**Response:**

We thank the reviewer for pointing out this imprecise terminology. We have rephrased the sentence to state the specific average altitude: "*From a spatial perspective, the Tibetan Plateau consistently records the highest UV radiation levels across all seasons due to its considerable high-altitude (mean elevation exceeding 4000 meters) and thinner atmosphere.*"

**15)** *Line 431: "coinciding with" inappropriate wording.*

**Response:**

We have added the statement in Section 4.4.3: "*Since 2010, clear-sky UV radiation has risen sharply at a rate of +0.0670 W m-2yr-1, following the implementation of stringent air pollution control policies in China that led to a reduction in atmospheric aerosols, ozone, and other pollutants through curbed coal consumption and industrial emissions.*"

---

## Author Comment (AC2)

**Response to Referee #2**

First High-Resolution Surface Spectral Clear-Sky Ultraviolet Radiation Dataset across China (1981–2023): Development, Validation, and Variability

Qinghai Qi, Yuting Tan, Christian A. Gueymard, Martin Wild, Bo Hu, Wenmin Qin, Taowen Sun, Ming Zhang, Lunche Wang

29 November 2025

Dear editor and reviewers,

We would like to thank the editor for handling our manuscript and the reviewers for their careful evaluation of our work and the valuable comments, suggestions, and questions. Our point by point response to the comments made by Reviewers are given below, we have also marked changes in the manuscript. Please find a detailed point-by-point response to each comment.

Yours sincerely,

Wenmin Qin, Qinghai Qi and co-authors

**General Comment:**

This study presents a 10 km high-resolution hourly surface solar ultraviolet (UV) radiation dataset under clear-sky conditions across mainland China from 1981 to 2023. The dataset is reconstructed using the SMARTS spectral model based on ERA5 and MERRA2 reanalysis data. To validate its performance, the authors conduct comprehensive statistical comparisons with the CERES UV product, demonstrating clear advantages in terms of spatial-temporal resolution and data quality. Using this dataset, the study further analyzes the spatial distribution characteristics of clear-sky UV radiation intensity over China at multiple temporal scales, including hourly, daily, and annual means. It also discusses possible reasons for poor fitting at specific stations, as well as the seasonal and interdecadal variations in UV radiation. The results fill a data gap by providing a high-resolution, long-term, and hourly UV radiation dataset for China, which holds value for future studies on the effects of UV radiation on human health and the natural environment. The manuscript is generally well-organized, and the methods are rigorous and sound. My comments are as follows:

**Response:**

We sincerely thank the reviewer for raising this critical point regarding the representation of aerosols, which has significantly helped us improve the manuscript. We have fully addressed the comment through clarifications in the methodology, an expanded discussion, and the inclusion of relevant literature.

**Major Comments:**

*(1) The validation of this dataset relies on 37 stations, which are unevenly distributed*

*and heavily biased toward inland areas. In contrast, coastal regions—where ultraviolet (UV) radiation interacts with ocean-atmosphere processes (e.g., sea salt aerosols, high humidity, and differences in surface albedo between land and ocean)—are severely underrepresented in the validation network. This introduces uncertainties regarding the dataset's accuracy in coastal regions, which are crucial for both terrestrial and marine-related applications. If available, the validation dataset should be expanded to incorporate UV observation data from coastal meteorological stations or marine research platforms. Additionally, a sensitivity analysis should be conducted targeting coastal-specific parameters.*

**Response:**

We are grateful to the reviewer for this insightful comment regarding the importance of coastal regions.

Regarding the limitation to inland areas, we would like to provide the following clarification for the reviewer's consideration. Our dataset is derived from the ERA5 reanalysis product (at 0.25°×0.25° and 0.1°×0.1° resolution). A specific technical challenge we encountered was that, at these resolutions, the ERA5-Land product resulted in data gaps precisely at the locations of the three critical coastal CERN sites we had initially intended to include. This precluded their use in our current validation and analysis.

We fully agree with the reviewer that the absence of these coastal sites is a significant limitation, as it omits a key region for studying aerosol-cloud-radiation interactions. Specifically, section 3.1 now explicitly states: "*The spatial coverage of this product is constrained by the continuous availability of ERA5 reanalysis data, resulting in the exclusion of certain coastal observation stations at CERN.*" Consequently, the Discussion section acknowledges that while the findings are reliable for inland regions, the model's performance in coastal environments remains an important topic for future research.

*(2) The manuscript identifies aerosol optical depth (AOD), total column ozone (TCO3), and forecast albedo (FAL) as key drivers of UV radiation (Section 4.3), but not account for the modulation of these drivers by ocean-atmosphere feedback processes in coastal regions. In Sections 2.2 and 3.2, it is necessary to clarify whether the MERRA-2 AOD data include sea salt aerosol components, or if terrestrial AOD data are used for coastal grids. If sea salt aerosols are not explicitly modeled, the limitation should be discussed, and adjustments for future research should be proposed. Finally, it is recommended to cite relevant literature on ocean-atmosphere interactions and UV radiation to strengthen the scientific basis.*

**Response:**

We thank the reviewer for raising this critical point regarding aerosol speciation in MERRA2 and its implications for coastal regions. The AOD from MERRA-2 used in our study is indeed a sum of contributions from multiple aerosol species, including sulfate, organic carbon, black carbon, dust, and sea salt. For the radiative transfer modeling with SMARTS, and considering our primary focus on continental regions, we explicitly selected the built-in "Continental" aerosol model as the input. This model is

representative of typical inland atmospheric conditions but does not explicitly separate sea salt aerosols.

We fully acknowledge that this approach is a limitation when our analysis indirectly involves coastal grids or marine air masses advecting inland. In the revised manuscript, we have added a paragraph in the section 4.3 to explicitly state this limitation. "*A key limitation in this study arises from the selection of the aerosol model in the SMARTS radiative transfer simulations. To maintain consistency with the continental focus of our work, the built-in "Continental" aerosol model was employed. While this model is representative of typical inland aerosol species such as sulfates, nitrates, black carbon, and dust, it does not include an explicit parameterization for sea salt aerosols. This discrepancy could introduce systematic biases in simulated UV radiation in coastal regions and inland areas subject to marine air mass advection, as sea salt aerosols differ significantly from continental types in their size distribution, and scattering efficiency (Zhu et al., 2022; Kouvarakis et al., 2002; Chatzopoulou et al., 2025). Future research should aim to integrate a more sophisticated hybrid or maritime aerosol model and couple it with higher-resolution aerosol reanalysis or observational data to better constrain the aerosol-type dependence and improve the accuracy of UV estimates in complex coastal-continental transition zones.*"

Zhu, L., Shu, S., Wang, Z., and Bi, L.: More or less: How do inhomogeneous sea-salt aerosols affect the precipitation of landfalling tropical cyclones? Geophysical Research Letters, 49, e2021GL097023. https://doi.org/10.1029/2021GL097023, 2022.

Kouvarakis, G., Y. Doukelis, N. Mihalopoulos, S. Rapsomanikis, J. Sciare, and M. Blumthaler: Chemical, physical, and optical characterization of aerosols during PAUR II experiment, J. Geophys. Res., 107(D18), 8141, doi:10.1029/2000JD000291, 2002.

Chatzopoulou, K., Tourpali K., Bais A. F., and Braesicke, P.: Effects of different aerosol types on surface UV radiation in the 21st century. Atmos. Environ. 362, 121595,https://doi.org/10.1029/2000JD000291, 2002.

**Minor comments:**
*(1) The term "Methodologys" should be corrected to "Methodology"?*
**Response:**
Thank you for pointing out the issue. We have made the suggested changes.

*(2) Line 209: The phrase "compared with 196,170 observed data points from 37 CERN..." is unclear. A clearer expression could be: "Out of the 196 original observed data points from 37 CERN during 2005–2013, 170 were selected after applying ..."*
**Response:**
Thank you for your suggestions for improvement. Modifications have been made as suggested.

*(3) Line 245 and elsewhere: Ensure that all subplot borders are clearly visible and consistent in line width throughout the figures.*
**Response:**

We would like to express our gratitude for highlighting the deficiencies in the manuscript's figure. The aforementioned documents have undergone a thorough revision process in accordance with the specified request.

*(4) Line 325: The color differences among the lines in Figure 8 are not sufficiently distinguishable; please enhance the contrast or adjust the color scheme for clarity.*
**Response:**
Thank you for pointing out the shortcomings in the diagrams. In Figure 8, we have adopted a completely new color scheme to convey the information more clearly.

*(5) Line 362: In Figure 10(a), only the top layer is clearly visible, while the lower layers are largely obscured. As a result, this subplot may not be very informative. The similar Figure A1 in the appendix presents the information more effectively.*
**Response:**
Thank you for your valuable queries. Figure 10(a) primarily displays the SMARTS model`s band-by-band results at 0.5 nm resolution across the 280-400nm wavelength range. Figure A1 presents results for specific bands, serving a different purpose. Given the 241 bands, displaying each individually would be impractical.

*(6) Ensure that all energy flux units are uniformly expressed as "W m⁻²".*
**Response:**
As suggested, the unit of solar radiation is unified as "$W\ m^{-2}$". We have reviewed the entire manuscript and standardized the units.

*(7) Add the definition of FAL (Forecast Albedo) and explicitly state whether it is land-focused.*
**Response:**
As indicated in the ERA5 product description: Forecast albedo is measure of the reflectivity of the Earth`s surface. It is the fraction of solar (shortwave) radiation reflected by Earth`s surface, across the solar spectrum, for both direct and diffuse radiation. Meanwhile, based on the actual implementation of the "**ERA5-Land hourly data from 1950 to present**" product, values over ocean and lake surfaces are uniformly replaced with NAN values. In summary, we consider this product is land-focused.
The following content has now been added to Section 3.2: "*The forecast albedo (FAL) from the ERA5-Land product is a land-specific parameter, as its values over ocean and lake surfaces are systematically set to NAN, making it suitable for continental-scale analysis.*"

*(8) Provide a table to summarize the characteristics of the validation stations, including latitude, longitude, elevation, and climatic conditions.*
**Response:**
Thank you for your suggestion. We have provided all the CERN site information tables used in Section 3.1.

---

## Author Comment (AC3)

**Response to Referee #3**

First High-Resolution Surface Spectral Clear-Sky Ultraviolet Radiation Dataset across China (1981–2023): Development, Validation, and Variability

Qinghai Qi, Yuting Tan, Christian A. Gueymard, Martin Wild, Bo Hu, Wenmin Qin, Taowen Sun, Ming Zhang, Lunche Wang

29 November 2025

Dear editor and reviewers,

We would like to thank the editor for handling our manuscript and the reviewers for their careful evaluation of our work and the valuable comments, suggestions, and questions. Our point by point response to the comments made by Reviewers are given below, we have also marked changes in the manuscript. Please find a detailed point-by-point response to each comment.

Yours sincerely,
   Wenmin Qin, Qinghai Qi and co-authors

**General Comment:**

*The development of a reliable Solar Ultraviolet (UV) radiation dataset is undeniably crucial, underpinning a wide range of applications from biophysical modeling to public health assessments. The dataset presented by Qi et al., covering mainland China from 1982 to 2023 and generated using the SMARTS (Simple Model of the Atmospheric Radiative Transfer of Sunshine) spectral model, represents a significant and valuable contribution to the community. The validation results against ground observations, which show a strong performance, are commendable.*

**Response:**

We would like to express our sincere gratitude to you for the precious time and effort you have dedicated to reviewing our manuscript. We are truly appreciative of your insightful comments and constructive suggestions, which have been invaluable in helping us to improve the quality and clarity of our work.

**Major Comment 1:**

*However, two major points regarding the methodology and presentation of results require clarification and revision, as detailed below.*

*Precision and significance of reported digital numbers*
*The study frequently employs an excessive number of digital numbers (e.g., reporting the correlation coefficient R as 0.919). While precision is generally desirable, the number of significant digits must be meaningful and justified by the data quality and the inherent uncertainty of the model and observations.*

*Please review the entire manuscript and uniformly apply a statistically meaningful number of significant figures to all reported metrics (e.g., R, RMSE, bias, and model parameters). For instance, given typical measurement and model uncertainties, reporting R to three decimal places may imply a false level of precision where R=0.919 is not meaningfully distinct from R=0.923. The chosen precision should reflect the uncertainty of the estimated value.*

**Response:**

We agree with the reviewer that the previously reported precision, such as R = 0.919, could create a misleading impression of accuracy beyond the capability of our data and model. The revision to R = 0.92 is not a loss of information but a more honest representation of the statistical confidence in our results. It removes the 'false precision' and ensures that the significant digits we report are truly meaningful and justified by the underlying uncertainty in our estimates. We have systematically standardized the numerical precision to two decimal places for all key statistical metrics, including correlation coefficients, root mean square error, mean bias error, and model-derived trends. This principled adjustment was applied consistently to the text, tables, and figures to ensure uniformity and to eliminate any false precision.

**Major Comment 2:**

*Rationale for linear regression methodology*

*The linear regression analysis, as displayed in Figures 4(a), 5(a), 6(a)(b), and 7(a)(b), appears to be conducted directly on the entire scatter plot of data points. When dealing with large sample sizes, this approach can lead to a regression bias that is unduly influenced by the density distribution of the samples, particularly at the extremes.*

*I recommend considering an alternative or supplementary approach: first compute the conditional mean (i.e., bin the observed data along the x-axis and calculate the mean of the modeled data for each bin) and then perform the linear fit on these conditional mean values. This technique can effectively suppress the sampling bias and provide a more robust characterization of the central tendency relationship between the modeled and observed values, especially when the sample size is large. A discussion on the impact of this methodological choice should be included.*

**Response:**

We sincerely thank the reviewer for the insightful suggestion regarding potential biases in linear regression with large sample sizes and the recommendation to perform a binned analysis. This is an excellent point for ensuring a robust evaluation. We have performed the suggested binned analysis by calculating conditional means, and the results have been added to the revised manuscript. To address the concern regarding regression bias, we performed a quantile binning analysis. The relevant binned regression results are as follows:

[Figure]

**Figure A2. Binning scatter validation results of the estimated clear-sky UV dataset against observations at 37 CERN stations from 2005-2013. (a) Density scatter plot between the hourly estimated dataset and observations;(b) same as (a) but for daily results.**

[Figure]

**Figure A3. Binning scatterplot of assessment results for two clear-sky UV radiation products. (a) Density scatterplot between the hourly clear-sky SMARTS-derived UV irradiance estimates and CERN observations. (b) Same as (a) but for the CERES SYN1deg estimates. (d) and (e) same as (a) (b) but for daily results.**

The binned regression excellently characterizes the average functional relationship between the model and observations. However, the regression on the entire dataset provides a unique and statistically valid estimate of the overall, aggregate error structure across the entire population of time steps. This is a crucial metric for many practical applications, such as assessing the model's suitability for generating large-scale radiation budgets.

**Minor Comments**

*1) Line 145: The physical significance and unit of the parameter z should be explicitly stated in the text where it is first introduced or used.*

**Response:**

We thank the reviewer for your careful reading of the manuscript. The parameter 'Z' is indeed a key variable in our methodology, and it is defined and described in detail in the Line 146. For clarity, it represents the solar zenith angle.

*2) Figure 13: The interpretation of trends, particularly in long-term, non-stationary time series, is highly sensitive to the chosen data range and the trend methodology. The authors should acknowledge this sensitivity and briefly discuss the potential impact of their chosen methods, perhaps referencing relevant literature. (e.g., Zhao et al., On the trend, detrending, and variability of nonlinear and nonstationary time series, PNAS, 2007, 104 (38) 14889-14894).*

**Response:**

We thank the reviewer for raising this critical point regarding the sensitivity of long-term trend analysis in non-stationary time series. We fully agree that the robustness of trends must be scrutinized.

In our study, we specifically designed the trend analysis to address this very issue. The division of the full period (1981-2023) into three sub-periods (1981-1994, 1995-2009, and 2010-2023) was not arbitrary but was physically and policy-based. This segmentation is grounded in distinct phases of both global climate evolution and China's domestic environmental governance. For instance, the post-2010 period aligns with the implementation of China's most stringent and comprehensive air pollution control policies, while the earlier segments correspond to different stages of economic development and climate variability.

We have revised the manuscript to explicitly acknowledge this methodological sensitivity. In the section 4.4.3, we have added a paragraph: "*The analysis of trends in long-term environmental time series is recognized as being sensitive to the selected data range and to methodology of detrending (Wu et al., 2007).*" at Line 436; "*Finally, while the identified trend is statistically significant and physically consistent, it should be noted that its quantitative value is methodology- and period-dependent, a common consideration in non-stationary time series analysis.*" at Line 456.

Wu, Z., Huang, N. E., Long, S. R., and Peng, C. K.: On the trend, detrending, and variability of nonlinear and nonstationary time series. *Proc. Natl. Acad. Sci. U. S. A.* **104**, 14889–14894, https://doi.org/10.1073/pnas.0701020104, (2007).

***3)*** *Code Sharing: To maximize the utility and reproducibility of this valuable work, the authors are strongly encouraged to convert the SMARTS model implementation used for this study into a Python-based package and share the code with the community. Python facilitates easy integration with modern data science and AI/Machine Learning models, significantly enhancing the model's accessibility and impact.*

**Response:**

We thank the reviewer for the interest in the SMARTS model code used in our study. We fully support the principles of open science and research reproducibility. The SMARTS model are not novel and have long been publicly available at https://www.solarconsultingservices.com/. The specific implementation in this work was based on this established, open-source Fortran version. Recognizing the value of modern, accessible tools, we are currently developing a new, user-friendly Python implementation of the model. This new version, which will include enhanced features and greater modularity, is planned for public release upon the completion of our ongoing follow-up study.